# Proteomic Approaches to Uncover Salt Stress Response Mechanisms in Crops

**DOI:** 10.3390/ijms24010518

**Published:** 2022-12-28

**Authors:** Rehana Kausar, Setsuko Komatsu

**Affiliations:** 1Department of Botany, University of Azad Jammu and Kashmir, Muzaffarabad 13100, Pakistan; 2Faculty of Environment and Information Sciences, Fukui University of Technology, Fukui 910-8505, Japan

**Keywords:** salt stress, proteomics, crops, antioxidants, reactive oxygen species, phytohormone

## Abstract

Salt stress is an unfavorable outcome of global climate change, adversely affecting crop growth and yield. It is the second-biggest abiotic factor damaging the morphological, physio-biochemical, and molecular processes during seed germination and plant development. Salt responses include modulation of hormonal biosynthesis, ionic homeostasis, the antioxidant defense system, and osmoprotectants to mitigate salt stress. Plants trigger salt-responsive genes, proteins, and metabolites to cope with the damaging effects of a high salt concentration. Enhancing salt tolerance among crop plants is direly needed for sustainable global agriculture. Novel protein markers, which are used for crop improvement against salt stress, are identified using proteomic techniques. As compared to single-technique approaches, the integration of genomic tools and exogenously applied chemicals offers great potential in addressing salt-stress-induced challenges. The interplay of salt-responsive proteins and genes is the missing key of salt tolerance. The development of salt-tolerant crop varieties can be achieved by integrated approaches encompassing proteomics, metabolomics, genomics, and genome-editing tools. In this review, the current information about the morphological, physiological, and molecular mechanisms of salt response/tolerance in crops is summarized. The significance of proteomic approaches to improve salt tolerance in various crops is highlighted, and an integrated omics approach to achieve global food security is discussed. Novel proteins that respond to salt stress are potential candidates for future breeding of salt tolerance.

## 1. Introduction

Agriculture is severely threatened by many problems arising from climate change and population growth [1]. Soil salinity is one of the factors contributing toward the decline of crop yield worldwide [2]. It results by the accumulation of water-soluble salts within soil layers above the threshold level, which adversely affects seedling growth and seed yield [3]. The occurrence of soil salinization depends on various anthropogenic activities, agricultural/farming practices, and soil type [4]. Saline soil consists of soluble salts, such as sulfates and chlorides of sodium, calcium, magnesium, or potassium. Nitrate, carbonate, and bicarbonate ions are also present in saline soil. The pH and exchangeable sodium percentage of saline soil are less than 8 and 15, respectively, while the electrical conductivity is more than 4 dS m^−1^ [5]. All types of soil with diverse physical, chemical, and biological properties are affected by salinization [6]. Globally, it is estimated that saline land is increasing due to weathering of indigenous rocks, low precipitation, irrigation with saline water, high surface evaporation, and poor farming practices, which would result in the salinization of 50% of all arable land by 2050 [7]. Proper mitigation strategies and a wide range of adaptations are required to cope with salinity stress for sustainable agriculture. The development of salt-tolerant varieties can help to overcome salinity stress.

Soil salinization is globally increasing due to the scarcity of rains and the increase in evapotranspiration, adversely affecting crop germination, growth, development, and yield [8]. The primary effects of salinity on crops are osmotic and ionic imbalances, eventually resulting in oxidative stress. However, the secondary effects include hormonal disruption and nutrient imbalances, ultimately leading to a reduction in photosynthesis and yield [9]. The cumulative action of these primary and secondary effects ultimately leads to increased complexity in protein formation involved in restoring or maintaining osmotic balance, ion homeostasis, and reactive oxygen species (ROS) scavenging [10]. The main target of salinity-induced changes in plants is the root system architecture and growth rate. The primary root response toward salinity stress is crucial for plant growth and development, which is facilitated by several phytohormones [11], such as abscisic acid (ABA) and auxins [12]. Plants build a response system against salinity stress by synthesizing, signaling, and metabolizing phytohormones via multiple levels of crosstalk [13]. However, salinity stress has complex effects on photosynthetic activity, depending on the cultivar and the duration/concentration of stress [14]. Photosynthetic activity is negatively influenced by salinity stress [15]. The value of crop quality and quantity can be enhanced by the use of bee pollination [16], improving global economic and dietary outcomes. The omics revolution allows the identification of genes and proteins involved in the acclimation, regulation, and adaptation of metabolic processes impacting photosynthesis and hormonal alterations under salinity stress [17].

Proteomics serves as the finest tool to identify the increasing number of proteins, which vary with distinct conditions or stresses in cells. Different proteomic studies have revealed the range of promising candidate proteins that regulate metabolic processes, hormonal crosstalk, and signaling pathways interpreting salinity tolerance in crops [18]. Proteins related to sugar metabolism, the antioxidant mechanism, and hormone signal transduction play a prime role in salt stress tolerance in crops. Additionally, soluble sugars are vital in maintaining protein stability as well as osmotic regulation. For example, the accumulated trehalose in rice (*Oryza sativa*) seedlings improved their salinity stress tolerance by efficient functioning of trehalose-phosphate phosphatase 3 [19]. Antioxidant proteins produced by crops remove ROS and helped in reducing the damage caused by salt stress. The enhanced activity of peroxidase conferred salt tolerance in a maize (*Zea mays*) inbred line [20]. ROS played a crucial role in providing osmotic protection and regulating the water balance in growing barley (*Hordeum vulgare*) sprouts. This enhanced the protective mechanisms by cell wall component synthesis and phosphatidyl-inositol signaling [21]. In this review, proteomic techniques used for the elucidation of the response and tolerance mechanism of salt stress in crops are summarized to reveal the potential associations between protein accumulation and metabolic homeostasis for stress acclimation. The potential application of integrated approaches, which involve advanced proteomics, genomics, and transcriptomics toward salt stress acclimation in crops, is discussed for better crop breeding to address food security challenges.

## 2. Morphological and Physiological Effects of Salt Stress on Crops

Plant morphology and physiology are constantly reshaped in response to triggering factors from the environment, either from air or from soil. Morphological parameters, such as shoot/root morphology, visible early senescence, and biomass of grains, are highly important for screening salt-tolerant crop cultivars [22]. At reproductive and grain stages, salt stress tolerance was explored among rice cultivars differing in degrees of vegetative-stage salinity tolerance. Yield and grain quality decreased among salt-susceptible cultivars, and each rice cultivar responded differently irrespective of the vegetative stage [23]. Salt tolerance in crops, which are rice, wheat (*Triticum aestivum*), soybean (*Glycine max*), maize (*Zea mays*), and sunflower (*Helianthus annuus*), depends on cultivar differences and efficient transition of metabolic processes involving hormonal crosstalk and osmotic adjustment (Table 1, Figure 1).

### 2.1. Rice

Salinity stress induced negative changes in cellular morphology and cell cycle progression, resulting in a prolonged growth duration and inhibitory effects on the seed germination and plant growth of rice [25]. The natural variation in salt tolerance among Australian rice wild relatives in comparison to selected cultivars has highlighted the potential of the exotic germplasm to provide new genetic variation for salinity tolerance [43]. The shoot length of the rice-tolerant cultivar decreased, while its fresh weight increased under salt stress for 1 week. The chlorophyll content and the Na/K ratio in the leaves was higher in the tolerant cultivar to better maintain photosynthesis and ion homeostasis [24]. A recent study on rice cultivars exposed to 50 mM NaCl demonstrated a tissue Na+ concentration of up to 600 mM [44]. Cell cultures act as model systems to investigate the salinity response, which can possibly mimic the plant response to stress. Higher activities of detoxifying enzymes, such as ascorbate peroxidase and catalase, were observed in the cell cultures of salt-tolerant rice [45]. In the salt-sensitive cultivar of rice, the shoot length/root fresh weight reduced under salt stress and metabolites, such as shikimate/quinate of the shikimate pathway, decreased in the leaves of both tolerant and sensitive rice cultivars [26]. Rice salt tolerance mainly depends on an efficient stress perception mechanism, which mediates control over ROS homeostasis via upregulation of scavenging enzymes and the ability to maintain a high Na/K ratio, as well as shikimate pathway metabolites.

Roots are the first salt-stress-responsive part of a plant. Salt stress inhibits proliferation and promotes cell expansion in the root apical meristem. Salt stress accumulated ABA in the primary root of rice, revealing that ABA plays an essential role in its growth [28]. Root border cells increased in the salt-tolerant variety, and the relative water content of the shoot/root decreased in the sensitive variety [27]. OsR3L1, which is one of the members of the hybrid proline-rich proteins gene family, synchronized salt tolerance in rice through regulation of gibberellic acid (GA), indole acetic acid, peroxidases, and apoplastic hydrogen-peroxide metabolism [46]. The fundamental role of ABA and hormonal crosstalk defines the root structure architecture in salt-stress-tolerant rice cultivars.

### 2.2. Wheat

The seed germination rate, germination potential, shoot length, and vigor index significantly decrease under salt stress. Physiological attributes, such as soluble sugars and malondialdehyde, increased and superoxide dismutase decreased in wheat seedlings under salt stress [30]. The flavonol content increased in a dose-dependent manner in wheat grains, with more intense purple-blue pigmentation after salt stress treatment. The intense-colored-grain wheat cultivars having a high anthocyanin content significantly maintained higher dry matter production and more than three times greater Na/K content [33]. The total fresh weight, shoot dry weight, and root dry weight/length decreased in wheat seedlings under salt stress with increased Na/K in the leaf [31]. The leaf aerenchyma and packing density of thylakoids increased along with ABA in the wheat leaf. Additionally, the concentrations of GA and jasmonic acid (JA) decreased under salt stress, while ABA increased in leaves [29]. The hormonal crosstalk, metabolic shift, anatomical adaptations, and reduced aerial growth are responsible for the survival of wheat under salt stress.

Plants benefit from extensive roots, which access deeper soil layers having lower salt concentrations under natural field conditions. The production of a well-developed root system under salt stress conditions is vital for above-ground biomass production [47]. The length and dry weight of roots reduced compared to the shoot in wheat under salt stress [48]. Salt stress led to a transient increase in NADPH oxidase activity, accompanied by the accumulation of hydrogen peroxide and proline in wheat roots [49]. Elevated malondialdehyde and accumulated ABA/JA with increased electrolyte leakage were observed in wheat roots, along with decreased GA/salicylic acid (SA) [29,32]. Reduction in the morphological attributes of roots under salt stress is an indication of oxidative stress caused by high Na ions in the vicinity. As a consequence of the high level of salinity, it is preferable to separate desirable compounds (proline and antioxidant enzymes) from undesirable ones (hydrogen peroxide and malondialdehyde). This separation requires compartmentalization in the cytosol. Hormonal crosstalk and metabolic modifications are essential for initial stress sensing, median stress acclimation, and final stress tolerance in wheat roots.

### 2.3. Soybean

In soybean, an active photosynthetic machinery is needed to cope with salt stress. High salt concentration augmented the levels of osmolytes (glycine betaine and proline), hydrogen peroxide, and malondialdehyde, as well as the activities of antioxidant enzymes (ascorbate peroxidase, catalase, superoxide dismutase, and peroxidase) in soybean leaves [36]. The maximal photochemical efficiency of photosystem II and photosystem I significantly decreased with the loss of reaction center proteins in the salt-sensitive soybean cultivar. Inversely, photosystem I reaction center protein abundance, stability, chloroplast ultrastructure, and leaf lipid peroxidation were not affected in the salt-tolerant cultivar [50]. Less ROS accumulation, higher antioxidant enzyme activity, and higher capacity of Na/K homeostasis were declared to be the key mechanisms of salt tolerance in roots of soybean-tolerant lines. Increased N acquisition and assimilation were beneficial for tolerant lines to accumulate amino acids, which contributed to osmotic regulation and N reserves. Finally, high tricarboxylic acid (TCA) cycle activity resulted in the production of organic acids, NADH, and ATP to support growth under salt stress [37]. Such physiological modifications/adaptations in leaf and root metabolism can lead toward the acquisition of salt tolerance in soybean.

Salinity stress is detrimental to root growth and extension, which ultimately leads to reduced seedling vigor. It significantly reduces soybean root length as well as nutrient uptake, chlorophyll content, soluble protein, sugar content, and biomass yield [34]. In soybean, the root length/volume with its fresh/dry weight decreased under salt stress [36]. The increase in the root epidermis and endodermis with the formation of lysogenic aerenchyma reveals the protective role to reduce Na influx and to minimize NADH with the uptake of toxic salt ions in soybean roots [35]. In salt-tolerant soybean, the primary root length, lateral roots, and biomass increased, as well as the activity of antioxidant enzymes [37]. The stable architecture of the root system is essential for shoot proliferation in crops under salt stress.

### 2.4. Other Crops

Salt stress drastically affected photosynthetic parameters and structural chloroplasts integrity with increased ROS content, promoting disturbance in the plant metabolism when compared to non-saline conditions [51]. The effective modulation of some metabolites, such as arabitol, glucose, asparagine, and tyrosine, contributed toward the maintenance of the osmotic balance and reduced oxidative stress in maize leaves [39]. In sunflower cultivars, different ionic concentrations increased in the leaf, along with a marked decrease in the stem diameter and biomass under salt stress [42]. Leaf proteins, ash, energy, dietary fiber, minerals, β-carotene, ascorbic acid, polyphenol content, flavonoid content, and total antioxidant capacity in leafy vegetables (*Amaranthus tricolor*) increased by salt stress [52]. Efficient modulation of photosynthesis and associated metabolites can help crops adapt to salt stress.

Salt stress increased the synthesis of secondary metabolites, such as alkaloids, saponin, and anthocyanin, in stress-exposed plants, while their production can be further enhanced via medium supplementation with plant growth promoters [53]. Cell wall characterization revealed that salt stress modulates the deposition of matrix polysaccharides, cellulose, and lignin in maize roots [54]. Salt stress reduced the feruloylation of arabinoxylans in the stem but increased the lignification of maize seedling roots [38]. In barley roots, the most statistically enriched biological pathway was phenyl-propanoid biosynthesis, along with an intense salt-induced lignin impregnation found only at the cell wall of the salt-tolerant cultivar [41]. For *Brassica rapa*, the root water content was seriously affected under salt stress and the contents of hydrogen peroxide and malondialdehyde increased in the salt-sensitive cultivar compared to the tolerant one [40]. The efficient adaptability of root metabolism and the process of lignification reflect the degree of salt tolerance in crops.

## 3. Proteomic Techniques to Analyze the Effect of Salt Stress on Crops

Understanding the salt stress responses of crops is essential for addressing yield losses and developing salt-tolerant cultivars. Label-free quantitative proteomic techniques have gained popularity in crop research related to salt stress. Differentially accumulated proteins involved in redox reactions, photosynthesis, salt responsiveness, and carbohydrate metabolism were identified using isobaric-tags for relative and absolute quantitation (iTRAQ)-based comparative protein quantification of two contrasting salt-responsive rice cultivars [55]. An iodoacetyl tandem mass tag (iodoTMT)-based proteomic approach was used to analyze *Brassica napus* seedlings under salt stress conditions. Sulfenylated sites were identified in differentially accumulated proteins mainly involved in photosynthesis and glycolysis [56]. Liquid chromatography–mass spectrometry (LC-MS) analysis was performed in the young seedlings of pigeon pea (*Cajanus cajan*), which identified differentially abundant proteins related to DNA binding with one finger transcription factor family and glycine betaine biosynthesis [57]. The salt response mechanism in crops, such as rice, wheat, soybean, barley, sweet potato (*Ipomoea batatas*), sugarcane (*Saccharum officinarum*), and chickpea (*Cicer arietinum*), has been explained more precisely by the application of various proteomic techniques (Table 2, Figure 2 and Figure 3).

### 3.1. Rice

Salt stress is a major constraint for rice cultivation globally. Both seedling and reproductive stages of rice are considered salt susceptible [77]. Using TMT-MS, mitochondrial ATPases and soluble N-ethylmale-imide-sensitive factor-attachment protein receptors were identified in the roots of the salt-tolerant cultivar [58]. In the salt-tolerant rice line, proteomic profiling identified differentially abundant proteins involved in stress/defense, and DNA replication/transcription accounted for the highest proportion, followed by protein transport and trafficking, carbohydrate metabolism, signal transduction, and cell structure [59]. Accumulation of mitochondrial and nuclear proteins resulted in salt stress acclimation in the roots of rice cultivars [58,59]. Increased proteins related to transcription and energy production are required to activate certain metabolic pathways for stress acclimation in crops.

High salinity severely restrains shoot growth and development, consequently leading to a reduction in grain yield. Proteins involved in photosynthesis and ROS scavenging accumulated in the shoots of the salt-tolerant rice variety compared to the sensitive one [60]. In rice leaves, the levels of catalase, peroxidases, and glutathione reductase were higher in sd58, which is a rice dwarf mutant, with enhanced salt tolerance than those in the wild type. This is consistent with the lower accumulation of ROS in sd58 than that in the wild type [61]. In rice grains, differentially accumulated proteins under salt stress conditions were mainly involved in the regulation of oxidative phosphorylation, phenylpropanoid biosynthesis, photosynthesis, posttranslational modifications, protein turnover, and energy metabolism [62]. Photosynthesis, energy, stress, ROS scavenging, and phenylpropanoid-related proteins play a pivotal role in rice shoots and grains for salt stress acclimation.

### 3.2. Wheat

Salt stress decreased seedling height, root length, relative water content, and chlorophyll content in wheat. Differentially accumulated proteins are mainly involved in oxidation-reduction, transmembrane transport, stress response, carbohydrate/carboxylic acid metabolism, and proteolysis [65,66]. Particularly, protein-disulfide isomerase and heat shock proteins increased and ribosomal proteins decreased [65]. MS-based proteomic analysis identified that proteins involved in the light-dependent reaction decreased under salt stress, while proteins associated with the Calvin cycle, plastoglobule development, protein folding/proteolysis, and hormone/vitamin synthesis increased. Salt stress is identified as more deleterious to wheat seedlings compared to osmotic stress [66]. Protein degradation, hormonal signaling, and efficient energy metabolism are needed for salt stress acclimation in wheat leaves.

The proteomic technique was used to identify the differentially abundant proteins from wheat roots in response to salt treatment using an absolute quantitation-based method. The differentially abundant proteins are ubiquitination-related proteins, transcription factors, pathogen-related proteins, membrane-intrinsic protein transporters, and antioxidant enzymes [64]. Wheat roots exhibited greater growth performance in response to salt stress as a result of increasing glutamine synthetase activity and nitrogen metabolism [78]. In another study of wheat, it was found that metabolism- and photosynthesis-related proteins increase, while proteins related to the TCA cycle decrease under salt stress [63]. The increased abundance of root aquaporins, late-embryogenesis-abundant proteins, dehydrins, plasma membrane Na/H antiporter, K transporters, and ABC transporters conferred stress tolerance in the salt-tolerant wheat cultivar [79]. These findings indicate that metabolic regulation triggered by membrane channel proteins/genes and hormonal signaling are the key tolerance mechanisms in wheat roots.

### 3.3. Soybean

Enzymes related to amino acid and glucose metabolism participated in the material decomposition during the soybean fermentation process under high salt stress [67]. The gamma-aminobutyric acid content and anti-oxidase activity increased in germinating seeds of soybean under salt stress [69]. In soybean, PM18 protein accumulated during radicle germination under salt stress. Its phosphorylated form was better able to protect lactate dehydrogenase from inactivation. During germination, the process of protein phosphorylation had regulatory effects on the stress-tolerance-related function of late-embryogenesis-abundant proteins [68]. Overexpression of GmMYB173 resulted in flavonoid accumulation in soybean transgenic roots under salt stress [70]. For better seed germination and seedling growth in soybean under salt stress, increased gamma-aminobutyric acid and PM18 proteins, improved amino acids, and carbohydrate metabolisms are critically important molecular strategies.

A comprehensive understanding of membrane lipid adaption following salt treatment was achieved by combining time-dependent lipidomic and proteomic data. Proteins involved in phosphoinositide synthesis and turnover increased at the onset of salt treatment in soybean leaves. Salinity-induced lipid recycling is shown to enhance JA and phosphoinositol production [72]. An iTRAQ-based proteomic technique was used to compare the abundance of proteins in untreated and salt-treated soybean leaves. Stress signal transduction, membrane proteins, ROS scavenging, protein metabolism, energy supply, and photosynthesis collectively functioned to reestablish cellular homeostasis under salt stress [71]. Efficient ROS scavenging, improved JA, enhanced phosphoinositol, and better lipid metabolism are the salt stress response mechanisms in soybean leaf.

### 3.4. Other Crops

Salt stress is one of the major devastating factors affecting the growth and yield of almost all crops, including important food crops, such as barley, sweet potato, and sugarcane. In germinating barley seeds, the protein profile was investigated using the TMT-based quantitative proteomic technique. Proteins associated with cellular redox homeostasis, the osmotic stress response, secondary metabolites derived primarily from amino acid metabolism, purine degradation, and shikimate pathways increased significantly in abundance. A high melatonin content in seeds was observed during germination [73]. To understand the tolerance mechanism toward salt stress, comparative protein analysis of salt-tolerant and salt-sensitive sweet potato roots was performed using tandem MS. It was found that significantly accumulated proteins are related to the regulation of ion accumulation, stress signaling, transcriptional regulation, redox reactions, plant hormone signal transduction, and secondary metabolite accumulation in the tolerant cultivar [74]. High melatonin, an enhanced shikimate pathway, redox homeostasis, and increased ionic/hormonal signal transduction confer salt tolerance in crop roots.

Protein quantification of sugarcane cultivars is performed with contrasting salt tolerance characteristics. Salt-tolerant sugarcane leaves show an upregulation of lipid-metabolizing enzymes, GDSL motif lipases, lipoxygenase, and type III chlorophyll a/b binding proteins as compared to the sensitive cultivar [75]. In chickpea leaves, salt tolerance was also studied using tolerant and sensitive cultivars. The accumulated proteins are chlorophyll a-b binding protein, oxygen-evolving enhancer protein, ATP synthase, carbonic anhydrase, ribulose 1,5-bisphosphate carboxylase/oxygenase (RuBisCO), heat shock and late embryogenesis abundant protein families, ascorbate peroxidase, elongation factor Tu, auxin-binding protein, and ribonucleoproteins [76]. The accumulated chlorophyll-related proteins, lipid-metabolizing enzymes, and stress/defense-related proteins play a pivotal role in the tolerance mechanism of crop leaves.

## 4. Roles of Proteomic Techniques for Improvement of Salt Tolerance of Crops

Proteomics refers to the study of protein characteristics at a high-throughput level, including protein expression level, post-translational modification, and protein–protein interaction [80]. Proteins related to the ROS-scavenging system and ABA activation are found exclusively in the salt-tolerant rice cultivar with higher auxin accumulation in roots [81]. Using the iTRAQ technique, differentially abundant proteins are found to be involved in phenylpropanoid biosynthesis, starch/sucrose metabolism, and mitogen-activated protein kinase signaling pathway in the salt-tolerant maize line. While in the sensitive line, differential proteins are associated with the nitrogen metabolism pathway [20]. In rice inoculated with ACC-deaminase-producing bacteria under salt stress, the accumulated proteins involved in antioxidant activities, RuBisCO, and ribosome are observed to enhance stress tolerance [82]. In the salt-tolerant alfalfa (*Medicago sativa*) cultivar, accumulated proteins are primarily enriched in the antioxidant system, starch/sucrose metabolism, and secondary metabolism; however, photosynthesis is inhibited in the sensitive cultivar due to the downregulation of the light-harvesting complex and photosystem-II-related proteins [83]. ABA activation, ROS scavenging, phenylpropanoid biosynthesis, and mitogen-activated protein kinase signaling are involved in the salt tolerance mechanism in crops.

Salt tolerance is regulated by a complex network of different component traits [84]. In rice, the over-expression of a ribosomal protein large subunit gene (RPL6) resulted in improved salt tolerance. An iTRAQ-based comparative proteomic analysis of leaves revealed that photosynthesis, ribosome/chloroplast biogenesis, ion transportation, transcription/translation regulation, and phytohormone and secondary metabolite signal transduction increased in a rice transgenic line [85]. Proteomic analysis of wheat leaves under salt stress revealed proteins mainly involved in stress defense, regulatory, protein folding/degradation, photosynthesis, carbohydrate metabolism, energy production, transportation, protein metabolism, and cell structure [86]. In the sugar beet (*Beta vulgaris*) monomeric addition line M14, differentially accumulated proteins were related to transport, metabolism, ROS homeostasis, stress/defense, biosynthesis, signal transduction, and transcription in roots under salt stress [87]. Photosynthesis, stress/defense, ribosomes, ROS homeostasis, and carbohydrate metabolism confer salt tolerance in transgenic lines of crops (Table 3).

### 4.1. Chemical Application

The improvement of salt tolerance in crops is indispensable to effectively dealing with food security challenges in different regions around the world [90]. Priming of plants with organic chemicals is a viable approach for the alleviation of salt effects in plants. The exogenous application of ethylene improves salt tolerance in barley plants. Matrix-assisted laser desorption–ionization-based protein identification results in high protein abundance of *S*-adenosylmethionine synthetase 3, which is a secretory protein located in the cell membrane and cytoplasm [89]. Enhanced ROS scavenging was observed in wheat seedlings treated with ethylene. The differentially accumulated proteins regulate protein localization, accumulation, folding, and degradation. Consequently, proteins involved in carbohydrate metabolism, cell wall synthesis, secondary metabolism, and the pentose phosphate pathway increase [88]. In wheat seedlings, exogenous JA increases salt tolerance by alleviating the antioxidant system; enhancing the contents of ABA, JA, SA; and regulating stress-related transcription factors [91]. Increased hormonal biosynthesis, the antioxidant system, and efficient carbohydrate metabolism are necessary for salt stress tolerance in crops after exogenous application of chemical substances.

### 4.2. Biotechnological Tool Genome Editing

Genome editing is a powerful technology to create new variations in the genome with desirable gene combinations. Methods, such as zinc finger nucleases (ZFNs), and transcription activator-like effector nucleases (TALENs), have been used previously [92]. Clustered regularly interspaced short palindromic repeats R-associated protein (CRISPR/Cas) technology is widely used for improving crop traits under salt stress by editing the activity of several protein-coding genes, and even miRNAs [93]. Genetic factors, which improve stress tolerance, are positive regulators, such as RAV2 encodes a transcription factor from the AP2/ERF family in rice. CRISPR-mediated deletion of GT-1, a promoter element in the RAV2 locus, confirmed its involvement in salt response [94]. Other examples of CRISPR-mediated targeting of kinase and phosphatase genes include FLN2 [95] and BBS1 [96] in rice. CRISPR/Cas9 has revolutionized crop research for salt-stress-related traits due to its ease of use, cost-effectiveness, multiplexing capability, and high efficiency.

Some genes function as negative regulators of the plant response to salt and other abiotic stresses. RR22, which encodes a type-B response regulator involved in cytokinin signaling, was knocked down using CRISPR/Cas9, thereby improving rice salt tolerance [97]. Transgene-free homozygous genetically engineered lines were obtained by T1 segregation, which is likely to offer the possibility of trait introgression in related varieties within a short time. The CRISPR-derived SPL10 mutant, a member of the SPL family, displays a salt-tolerant phenotype, suggesting that OsSPL10 is a negative regulator of rice response to salt stress [98]. CRISPR/Cas9 genome editing is used to create loss-of-function (knockout) or reduced-function (knockdown) mutants. These mutants are transgene free and may suffer less from regulatory concerns compared to transgenic plants.

### 4.3. Other Techniques

Next-generation sequencing-based genome-sequencing efforts have led to the decoding of the genome architecture, resulting in the development of a larger set of genomic resources, which has enabled the dissection of the underlying mechanisms or genetic basis for the functional characterization of several genes in diverse plant species [99]. In an elite Indian Basmati rice cultivar, enhanced seedling-stage salt-tolerance was developed through marker-assisted transfer of a major quantitative trait locus *Saltol* derived from a highly salt-tolerant line [100]. Three candidate salt-tolerant genes were identified in rice roots under salt treatment using genome-wide association studies and RNA-seq analysis [101]. Genomic information about superior haplotypes can help select parental lines with preferred alleles at each locus, which can then be integrated into breeding programs to custom-design crops with desired allelic combinations to develop superior pure lines or hybrids [102]. Whole-genome sequencing-based identification of genetic variation coupled with precise phenotypic data is used to perform genome-wide association studies. Collectively, such analyses have played a pivotal role in establishing marker trait associations and identification of superior alleles and haplotypes for several key traits in major crop plants, including rice, wheat, maize, chickpea, pigeon pea, and common bean (*Phaseolus vulgaris*) [103]. With the availability of sequencing-based trait mapping using a biparental population or germplasm, candidate genes for salt stress response were identified in chickpea [104]. Likewise, genes involved in auxin efflux and transportation were identified as salt- and anaerobic-stress-tolerant candidates in rice using genome-wide association analysis [105]. All these techniques focus on genome-sequencing data, and now there is a dire need to integrate these tools for efficient breeding of salt-tolerant crop varieties. Artificial-intelligence- and machine-learning-based computer science advances can provide rapid predictions of salinization processes from the agricultural field to the global scale [106]. The potential use of nitrogen-fixing legumes to restore saline soil is gaining attention and requires global evaluation [107]. Additionally, incorrect agricultural practices are causing great hindrance in gaining maximum crop yields. The adoption of effective cultivation practices in agriculture is direly needed.

## 5. Future Perspectives

Proteomics is composed of potent tools to elucidate salt-induced changes in the protein profile of crops. The integrated role of protein networks relevant to salt response could further explore tolerance mechanisms. The highly intricate physiological, biochemical, and molecular responses have made it difficult to comprehend salt tolerance mechanisms. Although many proteins and genes related to salt tolerance have been identified at the seedling stage from different crops, more effort is required to find out candidate genes at the reproductive and grain stages. The use of key physiological processes, such as ionic homeostasis and osmotic balance, should be considered while searching for novel candidate salt-tolerant genes. To gain a deeper insight into the mechanisms of salt tolerance in crops, an elaborative approach related to changes in proteins, metabolites, and genes is crucial. Biotechnological techniques, including overexpression of salt-responsive miRNAs, knockdown of their target genes using an RNAi approach, and genome editing using the CRISPR-Cas9 system, are the putative areas to be further explored. Candidate genes selected from RNAi knockdown studies can serve as new targets for future CRISPR-Cas9-based gene editing. Simultaneously, high-throughput phenotyping and genotyping of the crop germplasm to discover single-nucleotide polymorphisms and haplotype-based breeding approaches can be helpful in enhancing salt tolerance in crops. Adopting these new approaches would speed up the development of salt-tolerant crop varieties, which will ultimately reduce the threat to global food security (Figure 4).

## Figures and Tables

**Figure 1 ijms-24-00518-f001:**
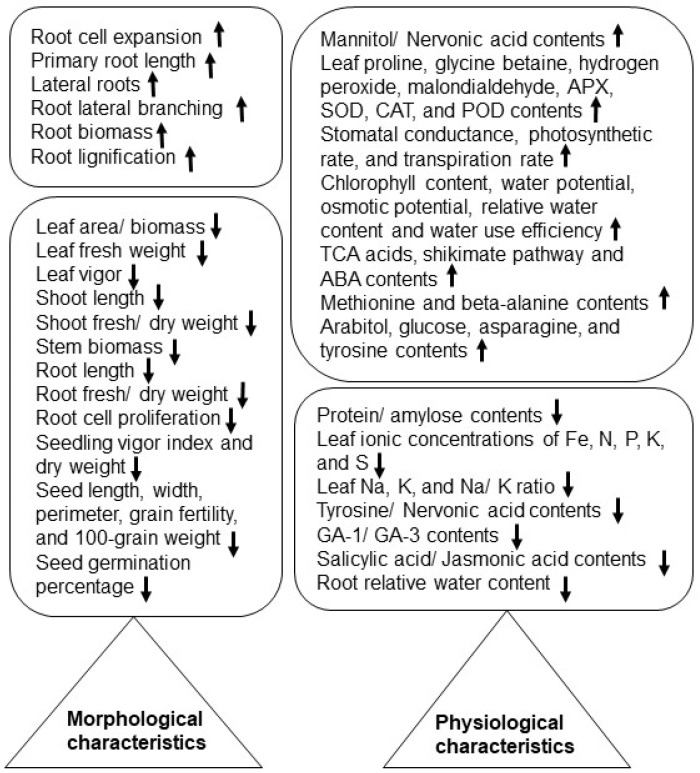
Morphological and physiological changes reported under salt stress in crops. Upward and downward arrows show an increase and decrease, respectively, in morphological or physiological characteristics.

**Figure 2 ijms-24-00518-f002:**
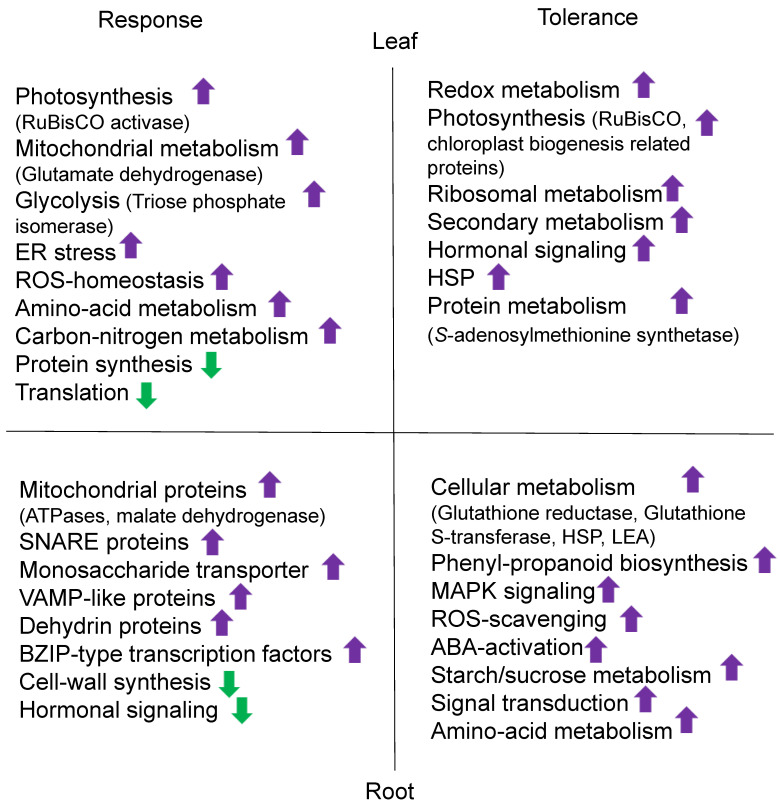
Salt stress response and tolerance mechanism in the leaves and roots of crops based on proteomics results. Upward and downward arrows show an increase and decrease, respectively, in morphological or physiological characteristics. Abbreviations: ABA, abscisic acid; ER, endoplasmic reticulum; ROS, reactive oxygen species; RuBisCO, ribulose bisphosphatase carboxylase/oxygenase; VAMP, vesicle-associated membrane protein; SNARE, soluble N-ethylmale-imide-sensitive factor-attachment protein receptors; HSP, heat shock protein; LEA, late embryogenesis abundant protein; BZIP, basic leucine zipper; MAPK, mitogen-activated protein kinase.

**Figure 3 ijms-24-00518-f003:**
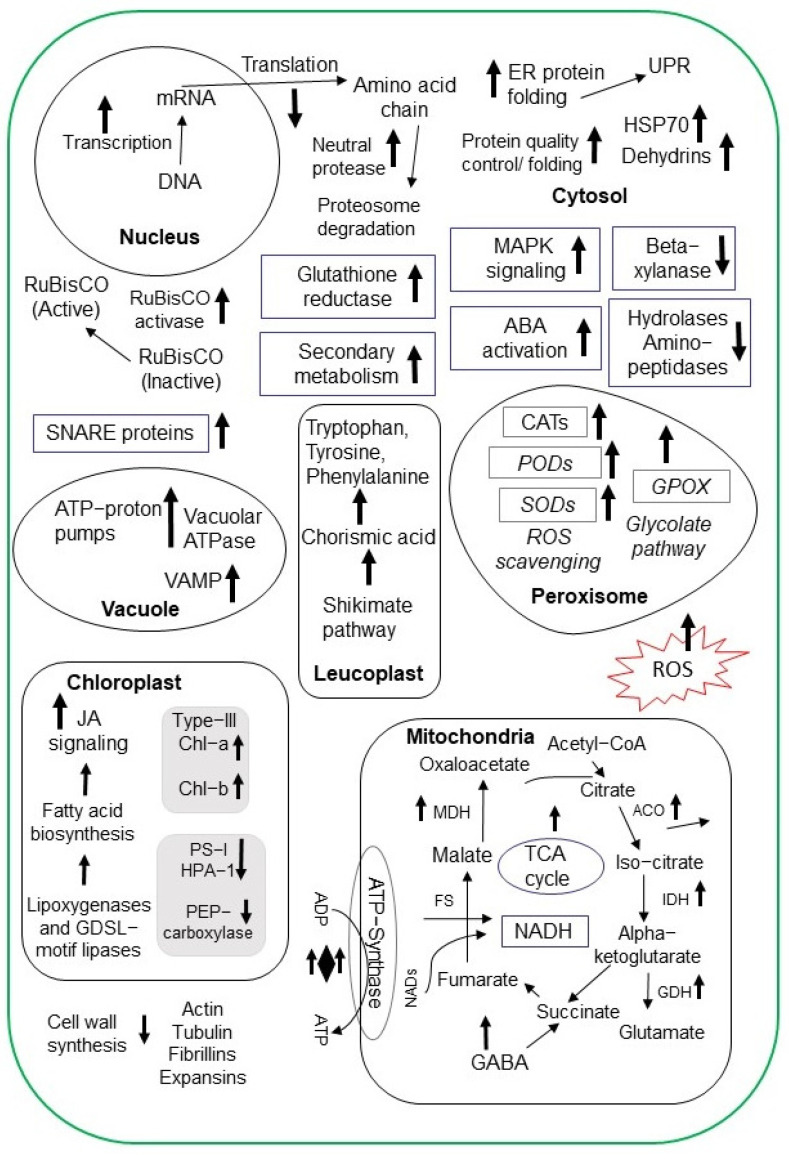
Proteomic alterations of salt-stress-responsive proteins in crop plants. Increased and decreased proteins are highlighted with upward and downward black-colored arrows, respectively. Salt-stress-induced changes in the proteins related to glycolysis, photosynthesis, the TCA cycle, and ROS production are represented. Under salt stress, major proteins belonging to the nucleus, chloroplast, mitochondria, peroxisomes, and endoplasmic reticulum are schematically presented. Abbreviations: HSP70, heat shock protein 70; ABA, abscisic acid; MAPK, mitogen-activated protein kinases; RuBisCO, ribulose-bisphosphate-carboxylase-oxygenase; CATs, catalases; PODs, peroxidases; SODs, super-oxide dismutases; GPOX, glycolate peroxidase; ER, endoplasmic reticulum; UPR, unfolded protein response; GABA, gamma-amino-butyric acid; VAMP, vesicle-associated membrane protein; acetyl-CoA, acetyl-coenzyme A; SNARE, soluble N-ethylmale-imide-sensitive factor-attachment-protein receptors; HPA1, harpin protein 1; Chl-a, chlorophyll a; Chl-b, chlorophyll b; PEP, phosphoenolpyruvate; GDSL, glycine, aspartic acid, serine, and leucine motif consensus amino acid; GDH, glutamate dehydrogenase; ACO, aconitase; IDH, isocitrate dehydrogenase; MDH, malate dehydrogenase; FS, fumarase; NADH, nicotinamide adenine dinucleotide hydrogen; NADs, nicotinamide adenine dinucleotides; ADP, adenosine diphosphate; ATP, adenosine triphosphate; TCA, tricarboxylic acid; ROS, reactive oxygen species.

**Figure 4 ijms-24-00518-f004:**
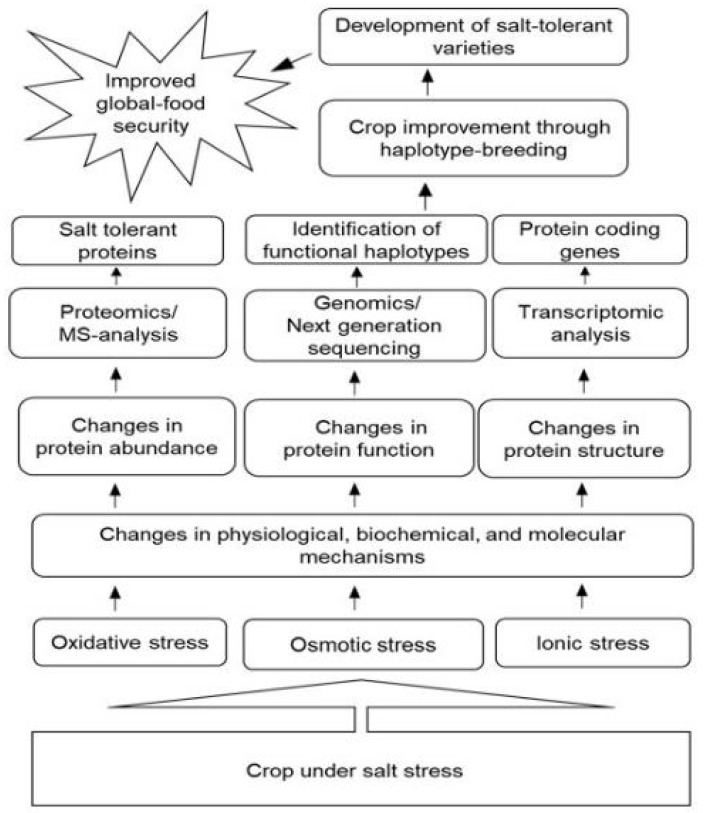
Integration of different omics techniques for crop improvement under salt stress conditions. A crop under salt stress faces oxidative, ionic, and osmotic stresses all together, which ultimately leads to changes in the physio-biochemical and molecular processes. These changes alter the structure, function, and abundance of proteins, which are the key players in cellular metabolism. The omics revolution has now made it possible to thoroughly study proteins and genes. Advanced genomics techniques allow the identification of candidate SNPs using NGS, followed by the identification of the closely linked adjacent SNPs within a specific range on the same chromosomal region (haplotype). Functional haplotypes can be used as an alternative breeding strategy for developing salt-tolerant crop varieties in a short span of time with a more targeted approach. Abbreviations: NGS, next-generation sequencing; SNPs, single-nucleotide polymorphisms.

**Table 1 ijms-24-00518-t001:** Morphological and physiological impacts and effects of salt stress on various crops studied during 2017–2022.

Crop	Organ	NaCl Conc. *	Morphological Effects	Physiological Effects	Metabolites	Ref. **
Rice	Shoot, leaf, root	60 mM	The shoot length, leaf area, shoot fresh weight, and root fresh weight decreased in the sensitive variety.	The Na/K ratio decreased in tolerant and sensitive varieties.	(-)	[24]
	Shoot, leaf, root	150 mM	The germination percentage, root length, and seedling dry weight reduced.	The stomatal conductance, photosynthetic rate, transpiration rate, chlorophyll content, water potential, osmotic potential, relative water content, and water use efficiency decreased.	Proline, malonaldehyde, and glutathione reductase increased.	[25]
	Leaf, root	50 mM	The shoot length, root weight. and root lateral branching decreased in the sensitive cultivar only.	In the leaf, most organic acids involved in TCA and the shikimate pathway decreased in sensitive and tolerant cultivars. In the root, mannitol increased in the tolerant cultivar.	Raffinose, fructose, ribose, sucrose, shikimate, and quinate decreased.	[26]
	Shoot, root	60 mM	The shoot fresh/dry weight and root dry weight decreased in tolerant and sensitive varieties. Root border cells increased in the tolerant variety.	The relative water content of the shoot and root decreased in the sensitive variety but was maintained in the tolerant one.	(-)	[27]
	Root	100 mM	In the root apical meristem, cell proliferation was inhibited and cell expansion was enhanced.	The ABA content accumulated in the primary root.	Ethylene increased.	[28]
	Grain	40 mM	The length, width, perimeter, and grain fertility/100-grain weight decreased in salt-sensitive varieties but was maintained in the salt-tolerant one.	Endosperm-starch accumulation was enhanced, and the protein/amylose content decreased in salt-susceptible varieties.	(-)	[23]
Wheat	Leaf, root	100 mM	The aerenchyma and packing density of thylakoids increased.	ABA and GA3/JA increased and decreased, respectively, in the leaf, but ABA/JA and GA1/SA increased and decreased, respectively, in the root.	Flavonoids increased.	[29]
	Seed	200 mM	The germination rate/index, mean germination time, and vigor index decreased.	Malondialdehyde and soluble sugars increased, and superoxide dismutase decreased.	Zeatin, anthocyanin, flavone, and flavonol increased.	[30]
	Seedling	100 mM	The total fresh weight, shoot dry weight, leaf chlorophyll, root dry weight, and root length decreased.	Ionic concentrations of Na, K, and Na/K decreased in the leaf.	(-)	[31]
	Root	100 mM		Proline/malondialdehyde contents increased along with the maximum electrolyte leakage.	NADPH oxidase increased.	[32]
	Shoot, leaf	200 mM	The shoot dry weight decreased in yellow/blue and increased in purple/dark-purple-colored grain cultivars.	The simple fluorescence ratio, flavonoid content, modified flavonoid index, proline content, and anthocyanins content increased.	The malondialdehyde content increased.	[33]
Soybean	Root	40/80 mM	The root length decreased.	Proline and glycine betaine increased.	Soluble sugars decreased.	[34]
	Stem, root	100 mM	The stem and root biomass decreased.	The pith and cortex increased at the internodes of the stem, and the root epidermis and endodermis increased with the formation of lysogenic aerenchyma.	(-)	[35]
	Leaf, stem, root	80 mM	The shoot length and fresh/dry weight, root length/volume, and root fresh/dry weight decreased.	Leaf proline, glycine betaine, hydrogen peroxide, malondialdehyde, APX, SOD, CAT, and POD increased. Leaf protein, soluble sugars, chlorophyll, phenols, flavonoids, DPPH, photosynthetic rate, stomatal conductance, and transpiration rate decreased.	Total phenolic and flavonoid contents decreased in the leaf.	[36]
	Seedling root	100 mM	The primary root length, lateral roots, and biomass increased in tolerant lines as compared to the sensitive one.	Antioxidant enzyme (SOD, APX, and CAT) activities, Na/K ratio, N content, and nitrogen use efficiency increased in salt-tolerant lines.	TCA, glyoxalate, and dicarboxylate metabolites increased.	[37]
Maize	Stem, root	100 mM	Root lignification increased.	Feruloylation of arabinoxylans in the stem decreased.	Ferulic acid increased	[38]
	Leaf	200 mM	The leaf fresh weight and biomass decreased.	Arabitol, glucose, asparagine, and tyrosine increased in the leaf.	Maltitol, raffinose, and cinnamic acid increased.	[39]
*Brassica rapa*	Root	150 mM	The root water content and root length decreased.	Hydrogen peroxide and malondialdehyde increased.	(-)	[40]
Barley	Seminal roots	100 mM	There was no significant change in the root length observed in sensitive and tolerant cultivars.	Tyrosine decreased and methionine and beta-alanine increased in the root tips of the sensitive cultivar. Nervonic acid increased and decreased in tolerant and sensitive cultivar root tips, respectively.	Amines increased, and stearic acid decreased.	[41]
Sunflower	Leaf, stem	100 mM	The leaf mass increased and stem mass decreased.	In the leaf, B, Cu, Zn, Mn, and Na ion concentrations increased. In the stem, Fe, N, P, K, S, and the K/Na ratio decreased.	Ionic concentrations of leaf Ca and Mg increased.	[42]

NaCl conc. *, NaCl concentration; Ref. **, reference; (-), not found; TCA, tricarboxylic acid; ABA, abscisic acid; GA, gibberellic acid; JA, jasmonic acid; SA, salicylic acid; APX, ascorbate peroxidase; SOD, superoxide dismutase; CAT, catalase; POD, peroxidase; DPPH, 2,2-diphenyl-1-picrylhydrazyl.

**Table 2 ijms-24-00518-t002:** Proteomic techniques used to analyze the salt response mechanism in crops studied during 2016–2022.

Crop	Organ	NaCl Conc. *	Proteomic Technique	No. **	Major Findings	Ref. ***
Rice	Root	80 mM	TMT-MS	200	Mitochondrial ATPases, SNARE proteins, monosaccharide transporter, and VAMP-like protein increased.	[58]
		150 mM	LC-MS/MS	178	The protein interaction network displayed connections between proteins involved in cell wall synthesis, transcription, translation, and defense.	[59]
	Shoot	200 mM	iTRAQ-LC-MS/MS	149	Glutamate dehydrogenase, RuBisCO activase, peroxidases, and triose phosphate isomerase increased.	[60]
	Leaf	150 mM	iTRAQ-LC-MS/MS	332	Photosynthesis and ROS homeostasis increased.	[61]
	Grain	150 mM	iTRAQ-LC-MS/MS	279	HPA1 decreased with impaired chlorophyll metabolism and photosynthesis.	[62]
Wheat	Root	50 mM	LC-MS/MS	41	SOD, malate dehydrogenases, dehydrin proteins, and V-ATPase increased.	[63]
		350 mM	LC-MS/MS	121	Substrate-recruiting E3 ubiquitin ligases increased BZIP-type transcription factor.	[64]
	Shoot	200 mM	LC-MS/MS	234	Protein folding, quality control, and ER stress-response-related proteins increased. Protein synthesis and translation-related proteins decreased.	[65]
		200 mM	LC-MS/MS	194	Chloroplast proteins involved in the light-dependent reaction, Calvin cycle, transcription, amino acid metabolism, and carbon/nitrogen metabolism increased.	[66]
Soybean	Soybean mash	18%	TMT-LC-MS/MS	42	Hydrolases, dipeptidase, neutral protease 2, leucine aminopeptidase, and beta xylanase decreased.	[67]
	Radicle	1.2 M	2DE-TripleTOF	12	Phosphorylated PM18 protein protected lactate dehydrogenase during salt stress acclimation.	[68]
	Germinating seeds	50 mM	LC-MS/MS	201	The GABA content and antioxidase activity increased.	[69]
	Root	200 mM	LC-MS/MS	412	Dihydroxy B-ring flavonoids increased as anti-oxidants and GmMYB173 was phosphorylated.	[70]
	Leaf	200 mM	iTRAQ-LC-MS/MS	278	Carbohydrate/energy metabolism, signaling, membrane/transport, stress/defense, protein synthesis, and redox homeostasis increased.	[71]
	Leaf	0.9%	LC-MS/MS	2049	Plastidial JA biosynthesis, phosphatidylinositol production, the TCA cycle, and the glycolysis pathway increased.	[72]
Barley	Germinating seeds	240 mM	TMT-LC MS/MS	68	Melatonin along with proteins related to cellular redox homeostasis, osmotic stress, secondary metabolites, purine degradation, and shikimate pathways increased.	[73]
Sweet potato	Root	150 mM	iTRAQ-LC-MS/MS	727	Proteins related to ion accumulation, stress signaling, transcriptional regulation, redox reactions, plant hormone signal transduction, and secondary metabolites increased.	[74]
Sugarcane	Leaf	160 mM	iTRAQ-LC MS/MS	189	GDSL-motif lipases, lipoxygenase, and type III chlorophyll a/b binding proteins increased, while phosphoenolpyruvate carboxylase decreased.	[75]
Chickpea	Leaf	100 mM	LC-MS/MS	364	Photosynthesis, bioenergy, stress responsiveness, protein synthesis/degradation, and gene transcription/replication increased in the tolerant cultivar.	[76]

NaCl conc. *, NaCl concentration; No. **, number of identified proteins; Ref. ***, reference; ATPases, adenosine triphosphatases; SNARE, soluble N-ethylmale-imide-sensitive factor-attachment protein receptors; V-ATPase, vacuolar proton-translocating adenosine triphosphatase; VAMP, vesicle-associated membrane protein; RuBisCO, ribulose bisphosphatase carboxylase/oxygenase; ROS, reactive oxygen species; HPA1, harpin protein 1; SOD, superoxide dismutase; BZIP, basic leucine zipper; ER, endoplasmic reticulum; GABA, gamma-aminobutyric acid; JA, jasmonic acid; GDSL, motif consensus amino acid sequence of glycine, aspartic acid, serine, and leucine around the active site serine; TCA, tricarboxylic acid; GmMYB, glycine max MYB proto-oncogene transcription factor; PM18, late embryogenesis abundant group 3 protein; TMT-MS, tandem mass tags–mass spectrometry; 2DE, two-dimensional electrophoresis; iTRAQ-LC-MS, isobaric tags for relative and absolute quantitation–liquid chromatography–mass spectrometry.

**Table 3 ijms-24-00518-t003:** Proteomic techniques used for the improvement of salt stress tolerance in crops studied during 2016–2022.

Crop	Organ	NaCl Conc. *	Proteomic Technique	No. **	Major Findings	Ref. ***
Rice	Root	100 mM	MALDI TOF/TOF MS/MS	27	The ROS-scavenging system and ABA activation increased in the tolerant cultivar.	[81]
	Seedling	200 mM	iTRAQ-LC-MS/MS	333	Photosynthesis, ribosome/chloroplast biogenesis, ion transportation, transcription/translation regulation, phytohormones, and secondary metabolite signal transduction increased in the transgenic line of RPL6.	[85]
	Leaf	150 mM	LC-MS/MS	160	Antioxidant proteins, RuBisCO, and ribosomal proteins increased.	[82]
Wheat	Leaf	160 mM	MALDI-TOF/TOF MS	81	Stress response/defense, regulatory, folding/assembly, and degradation-related proteins increased.	[86]
	Shoot, root	150 mM	iTRAQ/TMT-LC-MS/MS	1140	Carbohydrate metabolism, redox metabolism, cell wall, secondary metabolism, and the pentose phosphate pathway increased by ethylene treatment.	[88]
Maize	Root	180 mM	iTRAQ-LC-MS/MS	626	Proteins associated with phenylpropanoid biosynthesis, starch/sucrose metabolism, and mitogen-activated kinase signaling increased in the tolerant cultivar.	[20]
Sugar beet	Root	200 mM	IodoTMT-LC-MS/MS	462	Proteins involved in the regulation of ROS homeostasis, carbohydrate and amino acid metabolism, stress/defense, biosynthesis, and signal transduction increased.	[87]
Barley	Leaf	200 mM	MALDI-TOF/TOF MS	21	*S*-adenosylmethionine synthetase 3, photosynthesis, redox homeostasis, defense/stress, and primary metabolism increased in the salt-tolerant cultivar by exogenous ethylene application.	[89]
Alfalfa	Leaf	100 mM	iTRAQ-LC-MS/MS	438	In the tolerant cultivar, proteins involved in the antioxidant system, starch/sucrose metabolism, and secondary metabolism increased. In the sensitive cultivar, proteins related to the light-harvesting complex and photosystem II decreased.	[83]

NaCl conc. *, NaCl concentration; No. **, number of identified proteins; Ref. ***, reference; ROS, reactive oxygen species; ABA, abscisic acid; RPL6, ribosomal protein large subunit gene; RuBisCO, ribulose bisphosphatase carboxylase/oxygenase; MALDI-TOF, matrix assisted laser desorption ionization–time of flight; LC-MS, liquid chromatography–mass spectrometry.

## Data Availability

Not applicable.

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
