# Peer review of "Proteomic Approaches to Uncover Salt Stress Response Mechanisms in Crops"

_ijms, 2022, doi:10.3390/ijms24010518_

Round 1

Reviewer 1 Report

My review of the manuscript of “ Proteomic Approaches to Uncover Salt-Stress Response Mechanisms in Crops” (Manuscript ID: ijms-2047722) is as follows:

Dear Authors,

Salt stress is an increasingly threatening abiotic stress factors nowadays, threatening not only agriculture but our food supply globally. Although, the plant salt stress ressistance has not yet been fully explored. Using in sustainable manner the halotolerant plants and cultivars, we can decipher the role of growth, development, and defence mechanism under salt stress.

This review focusing on proteomic approaches and their mechanisms which capable and promoting the plants survival under stressful conditions. It can be highlighted (positively) that roughly up to 80% of the cited articles are from the last 3 years, which I expect this in the case of a review article. As a result, the authors summarize the most recently used methods and proteomic techniques. The article is well structured, there are few errors in it, that are not confusing (these are detailed below). In my view, it does not require a significant changes. However, I do not fully understand the reason, if soil salinity is such a big problem all over the world, why the authors hardly cite any European, American, African and Australian publications. Please extend this to these areas as well. What is the maximum salt concentration (in mM) that occurs in agriculture, or can it cause a problem? In some places, 200 and sometimes 350 mM concentration seems a lot to me. But, this can vary by plant species and even by variety, indeed. The scientific name of the species must be written once (at their first occurrence). Which species do the authors specifically mean by "Brassica"? On what basis are the more important species considered more important? Each of these species should be introduced briefly (even in the previous way). Explanations of abbreviations must be included in the Tables legends.

These methods and techniques are inevitably important and effective from the point of view of increasing the salt tolerance of plants, but at the end of the manuscript attention can also be drawn to a change of approach in terms of sustainability. (e.g. Incorrect agricultural cultivation requires a kind of change of attitude.) It may seem a bit far-fetched, but I think it is important to mention this here.

Based on these, and minor comments, I would suggest a major revision of the manuscript before publishing.

Line 176: missing point

L288: double space

L372: unnecessary parenthesis

L412: unnecessary letter 's'

L 432 Fig 2: I'm just wondering, I think this figure is not necessarily that simple, one-way. The reticular approach is more realistic. Of course, within reasonable limits.

L476: double space

Author Response

Reviewer 1

Salt stress is an increasingly threatening abiotic stress factors nowadays, threatening not only agriculture but our food supply globally. Although, the plant salt stress ressistance has not yet been fully explored. Using in sustainable manner the halotolerant plants and cultivars, we can decipher the role of growth, development, and defence mechanism under salt stress.

This review focusing on proteomic approaches and their mechanisms which capable and promoting the plants survival under stressful conditions. It can be highlighted (positively) that roughly up to 80% of the cited articles are from the last 3 years, which I expect this in the case of a review article. As a result, the authors summarize the most recently used methods and proteomic techniques. The article is well structured, there are few errors in it, that are not confusing (these are detailed below). In my view, it does not require a significant changes. 

However, I do not fully understand the reason, if soil salinity is such a big problem all over the world, why the authors hardly cite any European, American, African and Australian publications. Please extend this to these areas as well. 

Answer: We are sorry for this problem. The suggested publications have been cited in the revised manuscript as follows:

(1)Ondrasek, G.; Rathod, S.; Manohara, K. K.; Gireesh, C.; Anantha, M. S.; Sakhare, A. S.; Parmar, B.; Yadav, B. K.; Bandumula, N.; Raihan, F. Salt stress in plants and mitigation approaches. Plants 2022, 11, 717.

(2)Tarchoun, N.; Saadaoui, W.; Mezghani, N.; Pavli, O. I.; Falleh, H.; Petropoulos, S. A. The effects of salt stress on germination, seedling growth and biochemical responses of tunisian squash (Cucurbita maxima Duchesne) germplasm. Plants 2022, 11, 800.

(3)Yichie, Y.; Brien, C.; Berger, B.; Roberts, T. H.; Atwell, B. J. Salinity tolerance in Australian wild Oryza species varies widely and matches that observed in O. sativa. Rice 2018, 11, 66.

(4)Sandhu, D., Kaundal, A. (2018) Dynamics of salt tolerance: molecular perspectives. In: Gosal, S. S., Wani, S. H., editors. Biotechnologies of Crop Improvement. Volume 3. Cham, Switzerland: Springer International Publishing AG. p. 25-40.

(5)Abiala, A.; Abdelrahman, M.; Burritt, D. J.; Tran, L-S. P. Salt stress tolerance mechanisms and potential applications of legumes for sustainable reclamation of salt‐degraded soils. Land Degrad. Dev. 2018, 1–11.

What is the maximum salt concentration (in mM) that occurs in agriculture, or can it cause a problem? In some places, 200 and sometimes 350 mM concentration seems a lot to me. But, this can vary by plant species and even by variety, indeed. 

Answer: Thank you very much for your suggestion. The maximum salt concentration that occurs in agriculture is 600mM NaCl and crops cannot survive in this high salt concentration. 

As suggested, another publication has been cited in section “2.1 Rice”. Newly cited publication is as follows: “Isayenkov, S. V.; Maathuis, F. J. M. Plant salinity stress: Many unanswered questions remain. Front. Plant Sci. 2019, 10, 80”.

The scientific name of the species must be written once (at their first occurrence). Which species do the authors specifically mean by "Brassica"? On what basis are the more important species considered more important? Each of these species should be introduced briefly (even in the previous way). 

Answer: As suggested, the scientific names of all cited species have been incorporated in text with red color.

Explanations of abbreviations must be included in the Tables legends.

Answer: Explanations of abbreviations have been made accordingly in legends of tables 1, 2, and 3 highlighted in red color.   

These methods and techniques are inevitably important and effective from the point of view of increasing the salt tolerance of plants, but at the end of the manuscript attention can also be drawn to a change of approach in terms of sustainability. (e.g. Incorrect agricultural cultivation requires a kind of change of attitude.) It may seem a bit far-fetched, but I think it is important to mention this here.

Answer: The section of “4.3 Other techniques” in text has been corrected as follows: “Additionally, incorrect agricultural practices are causing great hindrance in gaining maximum crop yields. The adoption of effective cultivation practices in agriculture is direly needed.”

Based on these, and minor comments, I would suggest a major revision of the manuscript before publishing.

Answer: Thank you very much for your valuable comments. Based on comments from you and other two reviewers, the text has been corrected in red.

Line 176: missing point

Answer: The sentence at the end of section “2.3 Soybean” has been rephrased as follows: “In salt-tolerant soybeans, primary root length, lateral roots, and biomass increased beside the activities of antioxidant enzymes”.

L288: double space

Answer: Double space of L288 has been removed.

L372: unnecessary parenthesis

Answer: Unnecessary “and” has been removed as suggested.

L412: unnecessary letter 's'

Answer: We are sorry, there are many mistakes. It has been corrected.

L 432 Fig 2: I'm just wondering, I think this figure is not necessarily that simple, one-way. The reticular approach is more realistic. Of course, within reasonable limits.

Answer: Previous figure 2, which is new figure 4 has been updated as per suggestion.

L476: double space

Answer: Double space of L476 has been removed.

Reviewer 2 Report

In the manuscript, the authors investigate the effects of salinity on the morphological and physiological of crops. Moreover, the proteomics analysis of crops of study was determined. The study needs more clarification related to the effect of salt stress.  The manuscript can be accepted after minor revision.

Comments to authors:

1.     Mention the relationship between salinity of plants and its effects on the secondary metabolites.

2.     Show the effect of salt accumulation on the phytochemicals and the nutrition value of every plant.

3.     Add another column in the table explaining the change of metabolites.

4.     Add an illustration showing the morphological and physiological effects of salt stress on the crops you have mention

5.     What are the types of proteins detected from the proteomics analysis and their mode of action?

6.     Explain the effect of the detected proteins in metabolic profile of the crobs

7.     Please add some figure to the text of proteomics to improve the readability.

8.     Graphical abstract is highly recommended.

9.     Authors would add a sentence of conclusion in the abstract.

10.  Please add more key words

11.  Line 199- 201, rewrite please?

12.  The authors would unify the verb tense in the abstract.

13.  204, authors would enhance the quality of figure 1, and 2.

14.  Line 229-230, please add the references?

15.  Line 234, what does “sd58” stand for?

16.  Line 248-250, rewrite please.

17.  line 253, please mention the reference of this part.

18.  Authors could benefit from the following reference in the introduction:

Khalifa, S. A., Elshafiey, E. H., Shetaia, A. A., El-Wahed, A. A. A., Algethami, A. F., Musharraf, S. G., ... & El-Seedi, H. R. 2021: Overview of Bee Pollination and Its Economic Value for Crop Production. Insects, 12(8), 688.

Author Response

Reviewer 2

In the manuscript, the authors investigate the effects of salinity on the morphological and physiological of crops. Moreover, the proteomics analysis of crops of study was determined. The study needs more clarification related to the effect of salt stress.  The manuscript can be accepted after minor revision.

Comments to authors:

  1. Mention the relationship between salinity of plants and its effects on the secondary metabolites.

Answer: Thank you very much for your valuable suggestion. Additional reference has been quoted in text as follows: “Zahra, N.; Wahid, A.; Hafeez, M. B.; Lalarukh, I.; Batool, A.; Uzair, M.; El-Sheikh, M. A.; Alansi, S.; Kaushik, P. Effect of salinity and plant growth promoters on secondary metabolism and growth of milk thistle ecotypes. Life 2022, 12, 1530”.

  1. Show the effect of salt accumulation on the phytochemicals and the nutrition value of every plant.

Answer: As suggested, the relevant reference has been added in text as follows: “Sarker, U.; Islam, M. T.; Oba, S. Salinity stress accelerates nutrients, dietary fiber, minerals, phytochemicals and antioxidant activity in Amaranthus tricolor leaves. PLoS ONE 2018, 13, e0206388”.

  1. Add another column in the table explaining the change of metabolites.

Answer: As suggested, new column about metabolites has been added in table 1 in red.

  1. Add an illustration showing the morphological and physiological effects of salt stress on the crops you have mention

Answer: Thank you very much for your suggestion. New figure 1 has been prepared in text to illustrate morphological and physiological effects of salt stress on crop plants.

  1. What are the types of proteins detected from the proteomics analysis and their mode of action?

Answer: Proteins related to glycolysis, photosynthesis, tricarboxylic acid cycle, plasma membrane, and reactive-oxygen species scavenging have been identified from proteomic analysis. New figure 3 has been prepared.

  1. Explain the effect of the detected proteins in metabolic profile of the crops.

Answer: The detected proteins increased activities of antioxidant enzymes, Osmoprotectants, hormonal signaling, amino acids, and ionic concentrations. Activated pathways result in enhanced secondary metabolites to ameliorate stress in crops. New figure 1 has been prepared for the explanation of metabolic changes.    

  1. Please add some figure to the text of proteomics to improve the readability.

Answer: As suggested, new figure 3 has been included in text to explain protein changes in crops under salt stress.

  1. Graphical abstract is highly recommended.

Answer: When this article will be accepted, graphical abstract will be prepared for journal.

  1. Authors would add a sentence of conclusion in the abstract.

Answer: Thank you very much for your suggestion. Abstract has been updated in red as follows: “Novel proteins that respond to salt stress are potential candidates for future breeding of salt tolerance”.

  1. Please add more key words

Answer: As per suggestion, new key words “Antioxidants, Reactive-oxygen species; Phytohormone” have been added with red color in manuscript.

  1. Line 199- 201, rewrite please?

Answer: As per suggestion, text has been rephrased as follows: “Differentially accumulated proteins involved in redox reactions, photosynthesis, salt responsiveness, and carbohydrate metabolism were identified using isobaric-tags for relative and absolute quantitation (iTRAQ)-based comparative protein quantification of two contrasting salt-responsive rice cultivars”.

  1. The authors would unify the verb tense in the abstract.

Answer: We are sorry for this problem. The verb tense has been unified in the abstract in red. Corrected sentence has been marked in abstract with red color.  “Novel protein markers, which can be utilized for crop improvement against salt stress, are identified using proteomic techniques”.  

  1. 204, authors would enhance the quality of figure 1, and 2.

Answer: Resolutions of previous figures 1 and 2, which are figures 2 and 4, have been increased as per suggestion.  

  1. Line 229-230, please add the references?

Answer: Relevant references have been included in text with red color as follows: “Accumulation of mitochondrial and nuclear protein resulted in salt stress acclimation in the roots of rice cultivars [48, 49]”.  

  1. Line 234, what does “sd58” stand for?

Answer: This sentence has been corrected as follows: “In rice leaves, the levels of catalase, peroxidases, and glutathione reductase were higher in sd58, which is rice-dwarf mutant, with enhanced salt tolerance than those in wild type”.

  1. Line 248-250, rewrite please.

Answer: As per suggestion, text has been rewritten in manuscript with red color as follows: “Salt stress decreased seedling height, root length, relative water content, and chlorophyll content in wheat”.  

  1. line 253, please mention the reference of this part.

Answer: Relevant references 55 and 56 have been added in text highlighted in red color.

  1. Authors could benefit from the following reference in the introduction:

Khalifa, S. A., Elshafiey, E. H., Shetaia, A. A., El-Wahed, A. A. A., Algethami, A. F., Musharraf, S. G., ... & El-Seedi, H. R. 2021: Overview of Bee Pollination and Its Economic Value for Crop Production. Insects, 12(8), 688.

Answer: Thank you very much for your suggestion. This reference has been cited in introduction section as follows: “The value of crop quality and quantity can be enhanced by the use of bee pollination [Khalifa et al., 2021], improving global economic and dietary outcomes”.

Reviewer 3 Report

Article ID: ijms-2047722-peer-review-v1

Title: Proteomic Approaches to Uncover Salt-Stress Response Mechanisms in Crops

===================================================================

The review was well organized and performed and the theme is novel and very interesting, but some improvements are needed. Overall, the manuscript will meet the publishing standard of the journal after revisions.

Lines 35-37: Please review your information for this sentence. Is the soil considered saline if its electrical conductivity is less than 4 dS/m?

Lines 44-45: What is the difference between an ionic imbalance as a primary effect of salinity and a nutrient imbalance as a secondary effect of salinity?

Line 53: please cite the following review article:

Seif El-Yazal, S.A., Seif El-Yazal, M.A., Dwidar, E.F., Rady, M.M. (2015). Phytohormone Crosstalk Research: Cytokinin and its Crosstalk with Other Phytohormones – A review. Current Protein and Peptide Science, 16 (5): 395–405.

Lines 55-56: “Photosynthetic activity is negatively influenced by salinity stress.” Please add citation, such as:

Azzam, C.R., Zaki, S.-n.S., Bamagoos, A.A., Rady, M.M., Alharby, H.F. (2022). Soaking Maize Seeds in Zeatin-Type Cytokinin Biostimulators Improves Salt Tolerance by Enhancing the Antioxidant System and Photosynthetic Efficiency. Plants, 11(8): 1004.

Lines 56-58: Please add an appropriate citation.

Lines 145-148: As a consequence of the high level of salinity, it is preferable to separate the desirable compounds (proline, enzymes .........) from the undesirable ones (hydrogen peroxide, malondialdehyde, ...) each in a separate sentence.

Line 155: “Increased N acquisition and assimilation were beneficial for tolerant lines to accumulate ..........…”, should be “Increased N acquisition and assimilation was/is beneficial for tolerant lines to accumulate .............

Line 190-192: Please add an appropriate citation.

Lines 204-205: Please, the figure (1) needs to be more clear (increase the resolution).

Line 412: There is an extra letter “S”.

Line 433: Please, the figure (2) needs to be more clear (increase the resolution).

In general: The authors used the simple past (active or passive) for the parts of the review paper, although it is preferable to use the present simple.

Author Response

Reviewer 3

The review was well organized and performed and the theme is novel and very interesting, but some improvements are needed. Overall, the manuscript will meet the publishing standard of the journal after revisions.

Lines 35-37: Please review your information for this sentence. Is the soil considered saline if its electrical conductivity is less than 4 dS/m?

Answer: Thank you very much for your valuable suggestion. Saline soil has an electrical conductivity of more than 4 dS/m. The suggested correction has been incorporated in text in red as follows: “The pH and exchangeable sodium percentage of saline soil are less than 8 and 15, respectively while the electrical conductivity is more than 4 dS m-1”.

Lines 44-45: What is the difference between an ionic imbalance as a primary effect of salinity and a nutrient imbalance as a secondary effect of salinity?

Answer: Thank you very much for pointing out. During salinity stress, plants undergo excess salt concentrations leading towards ionic imbalance in cells as a primary effect. Later on, with the continuity of excess salt ions, nutrient imbalance takes place because of associated hormonal and metabolic changes as well as alterations in cellular development and gene expression.                                                                                                                                                                                                         

Text has been updated with red color as follows: “The primary effects of salinity in crops are osmotic and ionic imbalances, eventually resulting in oxidative stress. However, the secondary effects include hormonal disruption and nutrient imbalances ultimately leading to reduce photosynthesis and yield decline”.   

Line 53: please cite the following review article:

Seif El-Yazal, S.A., Seif El-Yazal, M.A., Dwidar, E.F., Rady, M.M. (2015). Phytohormone Crosstalk Research: Cytokinin and its Crosstalk with Other Phytohormones – A review. Current Protein and Peptide Science, 16 (5): 395–405.

Answer: Thank you very much for mentioning it. The suggested citation has been added in text as follows: “The primary root response towards salinity stress is crucial for plant growth and development, which is facilitated by several phytohormones [Seif El-Yazal et al., 2015]”.

Lines 55-56: “Photosynthetic activity is negatively influenced by salinity stress.” Please add citation, such as:

Azzam, C.R., Zaki, S.-n.S., Bamagoos, A.A., Rady, M.M., Alharby, H.F. (2022). Soaking Maize Seeds in Zeatin-Type Cytokinin Biostimulators Improves Salt Tolerance by Enhancing the Antioxidant System and Photosynthetic Efficiency. Plants, 11(8): 1004.

Answer: Thank you very much for referring, the suggested citation has been incorporated in text as follows: “Photosynthetic activity is negatively influenced by salinity stress [Azzam et al., 2022]”.

Lines 56-58: Please add an appropriate citation.

Answer: Relevant citation has been added in text in red as follows: “The omics revolution allows the identification of genes and proteins involved in the acclimation, regulation, and adaptation of metabolic processes impacting photosynthesis and hormonal alterations under salinity stress [Arif et al., 2020]”.  

Lines 145-148: As a consequence of the high level of salinity, it is preferable to separate the desirable compounds (proline, enzymes .........) from the undesirable ones (hydrogen peroxide, malondialdehyde, ...) each in a separate sentence.

Answer: The suggested corrections have been incorporated in text with red color as follows: “As a consequence of the high level of salinity, it is preferable to separate the desirable compounds (proline and antioxidant enzymes) from the undesirable ones (hydrogen peroxide and malondialdehyde). This separation requires compartmentalization in cytosol”.

Line 155: “Increased N acquisition and assimilation were beneficial for tolerant lines to accumulate ..........…”, should be “Increased N acquisition and assimilation was/is beneficial for tolerant lines to accumulate .............

Answer: Suggested correction in text has been incorporated in red as follows: “Increased N acquisition and assimilation was beneficial for tolerant lines to accumulate amino acids, which contributed to osmotic regulation and N reserve”.

Line 190-192: Please add an appropriate citation.

Answer: Relevant citation has been added in text in red as follows: “Cell-wall characterization revealed that salt stress modulated the deposition of matrix polysaccharides, cellulose, and lignin in maize roots [Zou et al., 2022]”.  

Lines 204-205: Please, the figure (1) needs to be more clear (increase the resolution).

Answer: As suggested, the resolution of previous figure 1, which is figure 2 has been increased.

Line 412: There is an extra letter “S”.

Answer: We are sorry for this mistake. Extra letter “s” has been removed.

Line 433: Please, the figure (2) needs to be more clear (increase the resolution).

Answer: As suggested, the resolution of previous figure 2, now figure 4 has been increased.

In general: The authors used the simple past (active or passive) for the parts of the review paper, although it is preferable to use the present simple.

Answer: Thank you very much for your valuable comment. Text has been corrected with red color using present tense.

Round 2

Reviewer 1 Report

The Authors have made the requested corrections, so I support the publication of the manuscript.